# FDG-PET/CT for Response Monitoring in Metastatic Breast Cancer: The Feasibility and Benefits of Applying PERCIST

**DOI:** 10.3390/diagnostics11040723

**Published:** 2021-04-19

**Authors:** Marianne Vogsen, Jakob Lykke Bülow, Lasse Ljungstrøm, Hjalte Rasmus Oltmann, Tural Asgharzadeh Alamdari, Mohammad Naghavi-Behzad, Poul-Erik Braad, Oke Gerke, Malene Grubbe Hildebrandt

**Affiliations:** 1Department of Nuclear Medicine, Odense University Hospital, DK-5000 Odense, Denmark; jakob.lykke.bulow2@rsyd.dk (J.L.B.); lasse.ljungstrom@rsyd.dk (L.L.); hjalte.rasmus.oltmann@rsyd.dk (H.R.O.); tural.asgharzadeh.alamdari@rsyd.dk (T.A.A.); mohammad.naghavi-behzad2@rsyd.dk (M.N.-B.); poul-erik.braad@rsyd.dk (P.-E.B.); oke.gerke@rsyd.dk (O.G.); malene.grubbe.hildebrandt@rsyd.dk (M.G.H.); 2Department of Oncology, Odense University Hospital, DK-5000 Odense, Denmark; 3Department of Clinical Research, University of Southern Denmark, DK-5000 Odense, Denmark; 4Open Patient Data Explorative Network, Odense University Hospital, DK-5000 Odense, Denmark; 5Centre for Personalized Response Monitoring in Oncology, Odense University Hospital, DK-5000 Odense, Denmark; 6Centre for Innovative Medical Technology, Odense University Hospital, DK-5000 Odense, Denmark

**Keywords:** response monitoring, metastatic breast cancer, PERCIST, SULpeak, visual assessment

## Abstract

Background: We aimed to examine the feasibility and potential benefit of applying PET Response Criteria in Solid Tumors (PERCIST) for response monitoring in metastatic breast cancer (MBC). Further, we introduced the nadir scan as a reference. Methods: Response monitoring FDG-PET/CT scans in 37 women with MBC were retrospectively screened for PERCIST standardization and measurability criteria. One-lesion PERCIST based on changes in SULpeak measurements of the hottest metastatic lesion was used for response categorization. The baseline (PERCIST_baseline_) and the nadir scan (PERCIST_nadir_) were used as references for PERCIST analyses. Results: Metastatic lesions were measurable according to PERCIST in 35 of 37 (94.7%) patients. PERCIST was applied in 150 follow-up scans, with progression more frequently reported by PERCIST_nadir_ (36%) than PERCIST_baseline_ (29.3%; *p* = 0.020). Reasons for progression were (a) more than 30% increase in SUL_peak_ of the hottest lesion (*n* = 7, 15.9%), (b) detection of new metastatic lesions (*n* = 28, 63.6%), or both (a) and (b) (*n* = 9, 20.5%). Conclusions: PERCIST, with the introduction of PERCIST_nadir_, allows a graphical interpretation of disease fluctuation that may be beneficial in clinical decision-making regarding potential earlier termination of non-effective toxic treatment. PERCIST seems feasible for response monitoring in MBC but prospective studies are needed to come this closer.

## 1. Introduction

Breast cancer is the most frequent type of cancer among women [1], and, despite improvements in primary breast cancer treatment, 20–30% of these women are diagnosed with metastatic breast cancer (MBC). MBC is an incurable disease with a need for life-long medical treatment and continuous response monitoring [2].

Various imaging methods and standardized criteria to assess treatment response have been proposed over the years. Today, the contrast-enhanced computed tomography (CE-CT) and the corresponding Response Evaluation Criteria in Solid Tumors (RECIST) 1.1 are recommended in clinical trials [3,4,5] while no specific recommendations are given for response monitoring in patients with MBC in current clinical guidelines [6]. The development of new metastases is the main challenge when monitoring patients with MBC, especially in the bones [7], which is a predilection site for metastases in breast cancer [8]. ^18^F-Fluorodeoxyglucose Positron Emission Tomography/Computed Tomography (FDG-PET/CT) has shown higher accuracy for the detection of bone metastases than CE-CT [9]. RECIST 1.1 is based on morphological changes in tumor lesions seen on CE-CT [3] although knowledge from FDG-PET/CT has been included in the revised RECIST 1.1 [3]. FDG-PET/CT has been introduced for clinical response monitoring in MBC sporadically in hospital settings as it may have the potential to better predict treatment response compared with CE-CT [10]. The PET Response Criteria in Solid Tumors (PERCIST) have been proposed as a standardized tool for treatment evaluation in solid tumors [11,12,13]. Sparse literature addresses the impact of RECIST 1.1 compared with PERCIST for response evaluation in MBC [14], but one retrospective study found a higher predictive value of progression-free and disease-specific survival for PERCIST compared with RECIST 1.1. [15]. Different approaches in PERCIST have been discussed, e.g., the one-lesion method defined by the highest standardized uptake values (SUV) normalized by lean body mass (SULpeak) in the hottest metastatic lesion, or the five-lesion method applying five target lesions as in RECIST 1.1. The two methods were compared in a previous study concerning the predictive value in patients with MBC, with no significant difference [16].

We have presented a case study of longitudinal response monitoring by PERCIST in a patient with MBC [17], and to our knowledge, no previous studies have addressed FDG-PET/CT and the PERCIST criteria for longitudinal response monitoring in patients with MBC.

While studies on PERCIST in a clinical setting are requested [18], we aimed to evaluate the feasibility and potential benefit of FDG-PET/CT and PERCIST one-lesion for longitudinal response monitoring in patients with MBC and compare the findings from PERCIST with visual assessment. Since PERCIST assessment enables identification of a nadir level of SULpeak, we explored the impact of introducing the nadir scan as a reference for response assessment.

## 2. Materials and Methods

A retrospective feasibility study was conducted at Odense University Hospital (Odense, Denmark) between September 2017 and September 2019. The study comprised women treated and monitored for MBC at the Departments of Oncology and Nuclear Medicine between September 2017 and December 2017. The institutional review board and the Data Protection Agency (Journal no. 17/29850) approved the study and all subjects signed a statement of consent. All procedures performed in studies involving human participants were following the ethical standards of the institutional and national research committee and with the 1964 Helsinki declaration and its later amendments.

The secure systems RedCap (Research Electronic Data Capture, Wanderbilt University, Nashville, TN, USA) and SharePoint (Microsoft, Redmond, WA, USA) have been used for data collection and data management.

### 2.1. Setting and Participants

Criteria for selecting the participants were: women receiving medical treatment for MBC, treatment monitoring by FDG-PET/CT, at least one baseline and one follow-up scan available for analysis, and signed statement of consent. Exclusion criteria were: incomplete medical records, a missing biopsy from a metastatic lesion (in patients with recurrent MBC), and the presence of other active cancers.

Data were extracted from medical records regarding patients’ age, time from primary breast cancer to MBC, histological characteristics of the primary tumor and metastatic lesion, and prior adjuvant treatment.

### 2.2. Imaging Techniques and Standardization Protocol

Rejection criteria by PERCIST were not enforced due to the retrospective study design [12]. However, during the entire period over which PET/CT scans were performed in this study, local FDG-PET/CT imaging guidelines followed the standards of the EANM procedure guidelines [19] for tumor imaging and we, therefore, expect the PERCIST criteria to be fulfilled. Exceptions to the local requirements were not documented, however, such incidents were rare. The local requirement on injection-to-scan-time was 60 ± 5 min and patients were required to be fasting for 6 h before PET/CT examination. Blood glucose levels were only measured regularly in patients with diabetes. FDG activity was administered intravenously in a dose of 4 MBq/kg body weight. All patients were scanned on General Electric (GE) Discovery PET/CT systems (GE Medical Systems, Milwaukee, USA) and patients who were not scanned consistently on the same scanner model with similar reconstruction settings on follow-up scans were rejected from the study.

PET/CT-scans were performed from the skull to the proximal femora using either generation 1 GE Discovery STE, VCT, or RX PET/CT systems or generation 2 with time-of-flight (TOF) GE Discovery 690 or MI PET/CT systems. Data were acquired with the following settings: CT for attenuation correction: 140 kV and 30–110 mA Smart mA, rotation time 0.8 sec, pitch 1.375:1, Noise Index 25, detector coverage 40 mm. Transverse images were reconstructed using filtered back projection with an attenuation kernel, slice thickness 3.75 mm, interval 3.27 or 2.79 mm on the Discovery MI. PET scans were performed in 3D with a scan time of 2.5 or 1.30 min/frame for the Discovery MI. Images were reconstructed iteratively using ordered subset expectation maximization (OSEM), 2 iterations, 21 or 28 subsets, slice thickness 3.27 or 2.79 mm specifically on the Discovery MI. On generation 2 systems reconstructions were performed using TOF and point-spread-reconstruction.

### 2.3. Visual Assessment of Scans

Data regarding baseline and follow-up FDG-PET/CT scans were retrospectively collected from medical records. The scan reports and response categorization had previously been qualitatively evaluated by altering specialists in Nuclear Medicine and were used for clinical decision-making at the Department of Oncology as part of daily clinical practice. Data regarding response categories for visual assessment were collected from the scan reports, and no criteria were applied for the visual assessment. Scan reports were blinded before the PERCIST evaluation.

### 2.4. Assessment of Comparability by PERCIST

Scans were screened for comparability according to the PERCIST criteria [11,12]. Scans were considered non-comparable if the matrix size at follow-up scans differed from that of baseline scan or if the mean standardized uptake value corrected for lean body mass (SULmean) in a reference volume of interest (reference VOI) in the liver or aorta differed more than 20% or 0.3 SUL units between the scans.

### 2.5. Assessment of Measurability by PERCIST

A lesion was considered measurable at the baseline scan if presenting a typical pattern of breast cancer metastases, and if the SULpeak of the lesion was greater than or equal to:1.5 × SUL_mean, liver_ + 2 × SD_liver_ (SD_liver_ = standard deviation of SUL_mean,liver_)

Metastatic lesions were detected by SULmax by automatic bookmarking, using PET-VCAR software suite 3.2 (Advantage Workstation, GE Healthcare, Chicago, IL, USA). Lesions were manually drawn using a fixed threshold iso-contour in cases where a lesion was not included in the automatic bookmarking. The SULpeak was calculated by the software.

### 2.6. Definition of PERCIST_nadir_

SULpeak of the single hottest metastatic lesion was registered for all eligible scans and analyzed according to the PERCIST practical guideline [11]. The nadir level of SULpeak (PERCIST_nadir_) was defined as the lowest level of SULpeak after the pre-treatment baseline during a treatment line.

PERCIST analyses were achieved by two researchers (JLB and LL) under the supervision of a senior Nuclear Medicine specialist (MGH), who did the initial drawing of VOI and typed the data.

### 2.7. Response Categorization by PERCIST

Response evaluation was categorized according to one-lesion PERCIST into complete metabolic response (CMR), partial metabolic response (PMR), stable metabolic disease (SMD), and progressive metabolic disease (PMD) using SULpeak of the baseline (PERCIST_baseline_) and PERCIST_nadir_ as references. PMD was defined as: an increase by >30% (>0.8 SUL units) in SULpeak between the baseline and follow-up scans, detection of new lesion typical of metastases, visually increased dissemination, or unequivocal progression in a non-target lesion. PMR was defined as a decrease of >30% in SULpeak (>0.8 SUL units). SMD was defined as an increase or decrease in SULpeak less than 30%. CMR was defined as a total resolution of FDG avidity in lesions when a lesion was considered indistinguishable from surrounding tissue, and if the SULpeak had decreased to less than that of the SUL in the liver. The PERCIST guideline [11] suggests measurement of the mean SULpeak in the anatomic location situated as close as possible to the site of the original tumor, but the SULpeak will differ significantly depending on the organ of involvement, and, therefore, the percentage change in cases of CMR was set at −100%. The first scan with detectable metastatic lesions after CMR was defined as progression as defined by the detection of new lesions. The SULpeak of this scan was then used as a reference for the subsequent measurement of SULpeak, and the percentage change was set to 0%. Categories of mixed or equivocal answers were not covered by the PERCIST.

## 3. Results

### 3.1. Patients Characteristics

Patients with MBC were identified at the Department of Oncology at Odense University Hospital. A patient flowchart has been shown in Figure 1. A total of 187 scans in 37 patients were eligible for analysis after the exclusion of 128 non-comparable scans. The median age at MBC diagnosis was 62.3 (45.8–73.5) years, and the median time from primary breast cancer until the diagnosis of metastatic disease was 7.10 (1.56–18.8) years. Characteristics of the primary tumors and the verifying biopsy from metastatic lesions are shown in Table 1 and Table 2, respectively. In total, two patients were diagnosed with primary advanced disease and hence, no biopsy from a metastatic lesion was performed due to the standard of care in our institution.

### 3.2. Applicability of PERCIST

PERCIST was applied to all comparable scans, and 35 of 37 (94.6%) patients had measurable disease according to PERCIST. In total, five patients had no measurable disease at the baseline scans, but in three cases a subsequent scan showed SULpeak exceeding the threshold of measurability as suggested by PERCIST. We were unable to detect any measurable lesions with PERCIST in two patients contributing with 12 scans. One of these patients had a lobular carcinoma and contributed with 10 scans. These two cases were kept for analysis to reflect daily clinical practice and as suggested by PERCIST [11], since measurable disease may appear over time when the disease mutates and progresses.

### 3.3. Agreement between PERCIST and Visual Assessment for Response Categorization

Response categories were assigned in 150 follow-up scans. Progression was deemed significantly more frequently by PERCIST_nadir_ (36%) than PERCIST_baseline_ (29.3%; *p* = 0.020), and visual assessment (23.3%; *p* < 0.001).

We found a moderate proportion of agreement (0.62, 95% CI 0.55—0.70) between response assessments by PERCIST_baseline_ and visual assessment (Table 3). The main discrepancies were noticed in cases where PERCIST_baseline_ suggested non-response (SMD or PMD), and the visual assessment suggested response (PMR) for the same cases.

Table 4 shows the agreement between response assessment by PERCIST_baseline_ and PERCIST_nadir_, and it shows moderate agreement between the two (0.85, 95% CI 0.79—0.91). PERCIST_nadir_ suggested non-response more often than PERCIST_baseline_.

### 3.4. Response Categories

The distribution of response categories in 150 follow-up scans was CMR 14% (*n* = 21), PMR 28% (*n* = 42), SMD 28.7% (*n* = 43), and PMD 29.3% (*n* = 44) when compared with PERCIST_baseline_ and CMR 14% (*n* = 21), PMR 20.7% (*n* = 31), SMD 29.3% (*n* = 44), and PMD 36% (*n* = 54) when compared with PERCIST_nadir_.

The waterfall plot in Figure 2 illustrates the response categories of PERCIST_baseline_ in the 150 follow-up scans. In 37 cases, new lesions were visualized as indicated by the dark-red bars. In 12 cases, new lesions were visualized despite a decrease in SULpeak of the hottest lesion.

Figure 3 illustrates the longitudinal response monitoring of one patient with 16 follow-up scans. Response assessment according to PERCIST of the follow-up scans were compared with the PERCIST_baseline_ and the PERCIST_nadir_, respectively. New lesions have been visualized by the dark-red bars. From the illustration, it can be seen for the fourth treatment line that PERCIST_nadir_ could detect progression earlier than PERCIST_baseline_.

## 4. Discussion

In this feasibility study, we found high applicability for PERCIST for longitudinal response monitoring in metastatic breast cancer with 95% of patients having measurable disease according to PERCIST. A moderate proportion of agreement was observed when comparing PERCIST to visual assessment in daily clinical practice, with PERCIST suggesting non-response more often than visual assessment. When introducing PERCIST_nadir_, the number of non-response categories increased further. Earlier detection of progression translates into treatment change and has potential patient benefit.

The lack of consensus for response criteria in FDG-PET/CT is due to the absence of studies applying PERCIST (and other criteria) in clinical settings [18]. To our knowledge, only one retrospective study has applied PERCIST compared with RECIST 1.1. in MBC patients [15]. They found the predictive value between baseline and first follow-up scan in favor of PERCIST. In the present study, we chose to apply the PERCIST one-lesion, since no significant differences have been found between one-lesion PERCIST and five-lesion PERCIST [16]. Women with MBC have in general very widespread disease and, therefore, the total lesion glycolysis (TLG), also suggested by PERCIST [12] seems unfeasible for daily clinical practice.

We found a moderate agreement between PERCIST and visual assessment. PERCIST_baseline_ seems more sensitive for the detection of progression compared with visual assessment, and the same holds for PERCIST_nadir_ compared with PERCIST_baseline_. Identifying progression earlier has the potential to lead to earlier discontinuation of treatment and subsequent prevention of toxicity of an ineffective treatment. However, on the other hand, considering a state as progressive too early also has the risk of discarding a treatment too soon.

The PERCIST [12] criteria suggest subsequent follow-up scans to be compared with the pretreatment baseline scan. In daily clinical practice, the previous scan is also often used as a reference to allow detection of progression, if the disease has regressed since baseline. However, using the previous scan as a reference implies a risk of missing continuously small increases of FDG-uptake on successive scans. The potential clinical impact of using the nadir scan instead of the baseline as the reference in an individual patient is illustrated in Figure 3. When the SULpeak decreases in cases of metabolic regression, comparison to the baseline scan would be misleading. It seems relevant to introduce the nadir level of SULpeak for monitoring such cancer lesions in PERCIST in the same way as it suggested for the nadir of the tumor size in RECIST 1.1 [3].

Using SULpeak for monitoring has the advantage of providing continuous values that can visualize the disease fluctuations and hence identify the nadir level. However, visualizing the percentage change of SULpeak, as suggested in Figure 2 and Figure 3, gave rise to considerations of the response category CMR. It seems misleading to measure the SULpeak of a lesion being indistinguishable from background activity since the SULpeak depends on the organ of involvement. CMR in a lung metastasis is suspected to have low SULpeak values compared with bone or liver metastases, and we suggest cases of CMR to be visualized as minus 100%. The SULpeak of the first metastatic lesion detected thereafter is then considered PMD. This will be in line with the detection of new lesions, which then should be the new reference for measurement of the coming changes in SULpeak.

Standardization of response monitoring patients with MBC may have advantages such as facilitating evidence-based treatment decisions and increasing agreement between observers. Despite increased time spent performing PERCIST analysis compared with visual assessment at the moment, the time is considered well spend since PERCIST has shown a better inter-observer variability [20] in MBC patients compared with visual assessment and is also suggested to be more reproducible [21]. Future software and automated analysis will have the potential to be time-sparing when using PERCIST in clinical settings. Further, standardized response evaluation could be of potentially great value if FDG-PET/CT should be introduced for future clinical trials where quantitative measurements are demanded.

A strength of this study is the representation of daily clinical practice for longitudinal response monitoring in MBC without the strict acquisition of the PERCIST standardization protocol. Further, no other studies have, to our knowledge, applied PERCIST for longitudinal response monitoring in MBC. However, due to the retrospective design of this feasibility study, we excluded a large number of scans because of non-comparability. Future studies and clinical practice will assumingly not have to exclude the same amount of scans, since the improvement in scanner techniques has resulted in less fluctuation in the SUV. Further, the standardization protocol suggested by PERCIST could not be fully documented, which must be taken into account when interpreting data [22]. Nevertheless, in our daily clinical practice, FDG-PET/CT followed the EANM guideline which to a large extent fulfills suggested standardization requirements by PERCIST, and non-comparable scans were excluded. Since this is a retrospective study, we must account for potential selection bias in the study group. No specific reason was given for which patients being monitored by FDG-PET/CT as opposed to CE-CT, which was the main alternative for response monitoring of MBC at our institution. Choosing response monitoring by FDG-PET/CT could be related to substantial tumor burden, more aggressive subtype, or preference by the individual oncologist. The relatively short inclusion period of three months may have resulted in missed patients with longer intervals between response monitoring scans due to a better overall response to treatment. Finally, the lack of a reference standard in the evaluation of progressive and non-progressive disease limits the study for validation of the response categories. Despite the limitations, we still consider it of great importance to report on the clinical benefits and shortcomings of PERCIST for response monitoring MBC patients.

Prospective studies with sufficient follow-up are needed to validate PERCIST in longitudinal response monitoring of MBC, preferably in large prospective studies where the criteria can be compared between studies. Alongside, future studies should aim at comparing one-lesion PERCIST, five-lesion PERCIST, and total lesion glycolysis to clarify the most optimal PERCIST approach for patients with MBC, who in general have a very widespread disease.

In conclusion, PERCIST provides a semi-quantitative response categorization with a useful variable, SULpeak, allowing monitoring of disease fluctuation and enabling identification of a nadir level of SULpeak. PERCIST seemed feasible for response monitoring in patients with metastatic breast cancer and may be beneficial in clinical decision-making regarding potential earlier termination of non-effective toxic treatment. PERCIST_nadir_ detected progression more frequently than visual assessment and PERCIST_baseline_. Prospective studies are needed to validate PERCIST in longitudinal response monitoring of metastatic breast cancer and to assess the influence of PERCIST on clinical decision-making.

## Figures and Tables

**Figure 1 diagnostics-11-00723-f001:**
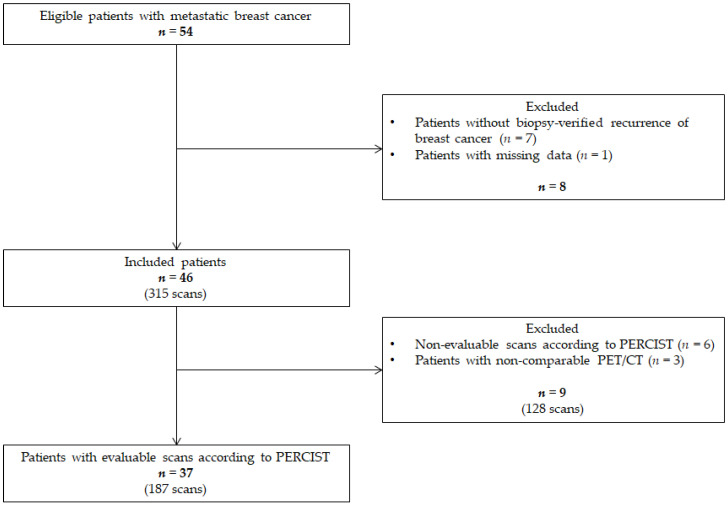
Flowchart of 109 patients with metastatic breast cancer screened for inclusion in the Department of Oncology, Odense University Hospital, from September 2017–December 2017. MRI magnetic resonance imaging, CT: computed tomography, PET/CT: positron emission tomography with integrated computed tomography, PERCIST: PET Response Criteria in Solid Tumors.

**Figure 2 diagnostics-11-00723-f002:**
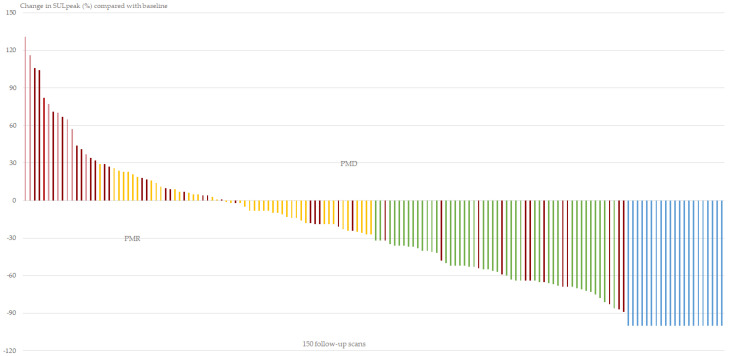
A waterfall plot with the percentage change in SULpeak in 150 follow-up scans compared with the baseline standardized uptake values normalized by lean body mass (SULpeak). In 37 cases new lesions were visualized by the dark-red color. In 12 cases, new lesions were visualized despite the favorable partial metabolic response category. Complete metabolic response indicated by blue bars was assessed visually. Response categories: PMD (**light-red**): PMD: progressive metabolic disease, SMD (**yellow**): stable metabolic disease, PMR (**green**): partial metabolic response, CMR (**blue**): complete metabolic response.

**Figure 3 diagnostics-11-00723-f003:**
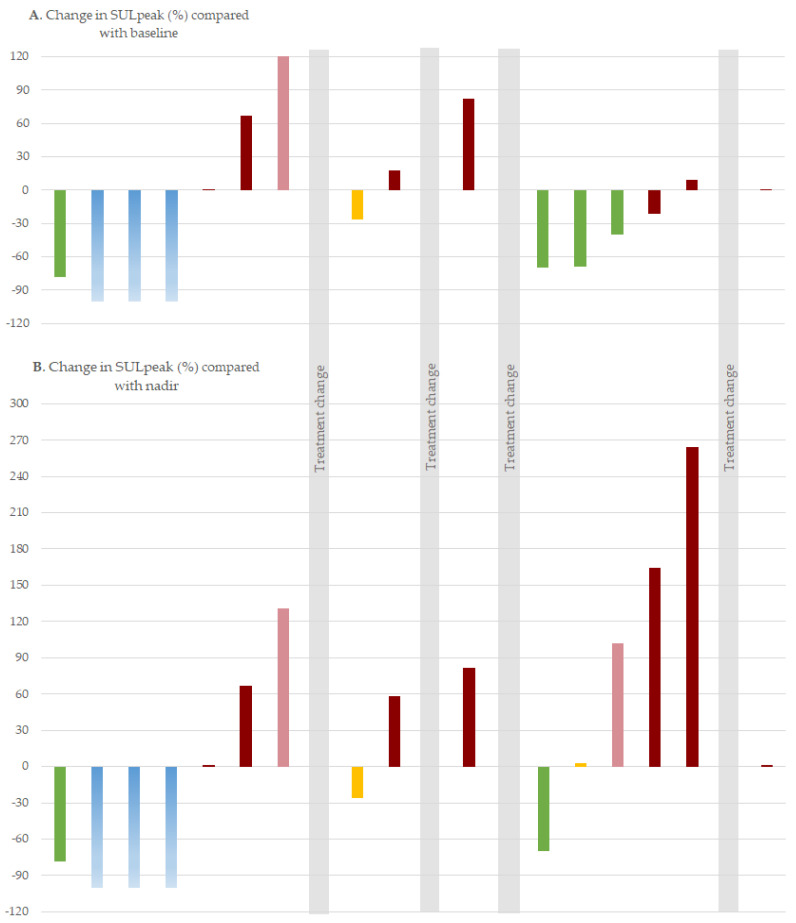
Illustration of the percentage change in standardized uptake values normalized by lean body mass (SULpeak) for one patient with 16 follow-up scans for (**A**) PERCIST response assessment compared with baseline and (**B**) PERCIST compared with the nadir level of SULpeak. New lesions were visualized by dark-red bars and change of treatment by shaded grey bars Response categories: PMD (**light-red**): Progressive metabolic disease, SMD (**yellow**): Stable metabolic disease, PMR (**green**): Partial metabolic response, CMR (**blue**): Complete metabolic response.

**Table 1 diagnostics-11-00723-t001:** Characteristics of primary breast cancer in 37 patients with metastatic breast cancer.

		*n* (%)
Type of Surgery	Breast conserving	11 (29.7)
	Mastectomy	21 (56.8)
	No surgery	4 (10.8)
	Other	1 (2.70)
Histology	Invasive ductal carcinoma	26 (70.3)
	Invasive lobular carcinoma	3 (8.11)
	Invasive carcinoma unspecified	5 (13.5)
	Unknown	3 (8.11)
Size	≤10 mm	2 (5.41)
	11–20 mm	11 (29.7)
	21–50 mm	11 (29.7)
	≥50 mm	6 (16.2)
	≤10 mm	2 (5.41)
Lymph node involvement	0 or micro-metastases	9 (24.3)
	1–3	10 (27.0)
	4–9	7 (18.9)
	≥10	3 (8.11)
	Unknown or no surgery	8 (21.6)
Grade	I	5 (13.5)
	II	11 (29.7)
	III	9 (24.3)
	Unknown	12 (32.4)
ER-status	Positive	29 (78.4)
	Negative	4 (10.8)
	Unknown	4 (10.8)
HER2-status	Positive	9 (24.3)
	Normal	18 (48.6)
	Unknown	10 (27.0)
Medical treatment	Neo-adjuvant +/− adjuvant	9 (24.3)
	Adjuvant only	22 (59.5)
	No medical treatment	6 (16.2)
		37 (100)

ER: Estrogen receptor, HER2: Human Epidermal Growth Receptor 2.

**Table 2 diagnostics-11-00723-t002:** Biomarker profile of metastatic lesion and location of biopsy in 37 patients with metastatic breast cancer.

		*n* (%)
ER-status	Positive	30 (81.1)
	Negative	6 (16.2)
	Unknown	1 (2.70)
HER2-status	Positive	9 (24.3)
	Normal	23 (62.2)
	Unknown	5 (13.5)
Location of biopsy *	Bone	11 (29.7)
	Lymph node	9 (24.3)
	Liver	6 (16.2)
	Other	4 (10.8)
	Breast	2 (5.40)
	Lung	2 (5.40)
	Skin	2 (5.40)
	Brain	1 (2.70)
		37 (100)

* Location of biopsy from a metastatic lesion, ER: Estrogen receptor, HER2: Human Epidermal Growth Receptor 2.

**Table 3 diagnostics-11-00723-t003:** The association between response assessment by visual assessment and PERCIST in 150 follow-up scans in patients with metastatic breast cancer.

**Visual Assessment**	**PERCIST**					
		**CMR**	**PMR**	**SMD**	**PMD**	**Sum**
	CMR	15	0	0	0	15
	PMR	5	37	22	11	75
	SMD	0	4	13	2	19
	PMD	0	0	7	28	35
	Mixed Response	0	0	1	1	2
	Equivocal Answer	1	1	0	2	4
	Sum	21	42	43	44	150
	Agreement	Expected agreement	Kappa	Std. Error	Z	Prob > Z
	62%	25.9%	0.49	0.04	11.0	<0.0001
	Overall agreement: 0.62 [95% CI 0.54–0.70]				Standard Error: 0.04	

PERCIST: PET Response Criteria in Solid Tumors, CMR: Complete metabolic response, PMR: Partial metabolic response, SMD: Stable metabolic disease, PMD: Progressive metabolic disease, Std.Error: Standard Error, CI: Confidence interval.

**Table 4 diagnostics-11-00723-t004:** The association between response assessment by PERCIST in baseline (PERCIST_baseline_) and nadir (PERCIST_nadir_) scans in 150 follow-up scans in patients with metastatic breast cancer.

	PERCIST_nadir_					
		CMR	PMR	SMD	PMD	Sum
PERCIST_baseline_	CMR	21	0	0	0	21
	PMR	0	29	10	3	42
	SMD	0	2	34	7	43
	PMD	0	0	0	44	44
	Sum	21	31	44	54	150
	Agreement	Expected agreement	Kappa	Std. Error	Z	Prob > Z
	85.3%	26.7%	0.79	0.048	16.6	<0.0001
	Overall agreement 0.85 [95% CI 0.79–0.91]				Standard Error: 0.03	

PERCIST: PET Response Criteria in Solid Tumors, CMR: Complete metabolic response, PMR: Partial metabolic response, SMD: Stable metabolic disease, PMD: Progressive metabolic disease, Std.Error: Standard Error, CI: Confidence interval.

## Data Availability

The data presented in this study are available on request from the corresponding author. The data are not publicly available due to private and ethical reasons.

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
