# Peer review of "FDG-PET/CT for Response Monitoring in Metastatic Breast Cancer: The Feasibility and Benefits of Applying PERCIST"

_diagnostics, 2021, doi:10.3390/diagnostics11040723_

Round 1

Reviewer 1 Report

the manuscript is worthwile, a few comments

  • since the correlation with the qualitative judgment is quite good, one can wonder if applying PERCIST adds much, as it is laborintensive to perform
  • i think it is important to mention that stopping a treatment earlier, possibly can prevent toxicity of an ineffective treatment on the one hand, however, on the other hand, considering a state as progressive disease too early also has the risk of discarding a treatment too soon. this is also an important risk, worth to mention.
  • the fact that using the nadir as a reference results in PMD earlier than using baseline as a reference is quite obvious, and perhaps does not need statistical testing.

Author Response

Referee 1

The manuscript is worthwile, a few comments

Comment 1

Since the correlation with the qualitative judgment is quite good, one can wonder if applying PERCIST adds much, as it is laborintensive to perform

Reply: We acknowledge that PERCIST is more laborintensive compared with visual/qualitative assessment at the moment. However, no studies have addressed the clinical utility of PERCIST for longitudinal response monitoring and clinical validation seems of great importance. PERCIST has the potential to help and guide clinicians with potential time-saving automated analysis if the software for this is improved. Further, the clinical benefits of PERCIST for response monitoring MBC patients is that it provides a semi-quantitative response categorization with a useful variable, SULpeak, allowing monitoring of disease fluctuation and enabling identification of a nadir level of SULpeak. This could further be of advantage for introducing FDG-PET/CT in clinical trials, where quantitative measurements are needed for patients to be enrolled in pharma studies.

We have elaborated on this in the discussion section.

Comment 2

I think it is important to mention that stopping a treatment earlier, possibly can prevent toxicity of an ineffective treatment on the one hand, however, on the other hand, considering a state as progressive disease too early also has the risk of discarding a treatment too soon. this is also an important risk, worth to mention.

Reply: We agree on this relevant point and have added this in the discussion section.

Comment 3

The fact that using the nadir as a reference results in PMD earlier than using baseline as a reference is quite obvious, and perhaps does not need statistical testing.

Reply: We agree on this point, however, we prefer to keep the statistical testing to show the difference.

Marianne Vogsen

Reviewer 2 Report

The paper addresses an interesting topic, as breast cancer, especially at an advanced stage, is a serious healthcare problem and monitoring response to treatment could provide fundamental support to clinicians in addressing the best treatment options and thereby improving patients' survival. Nevertheless, in my opinion the sentence "Not all standardization criteria suggested by PERCIST could be documented due to the retrospective study design [12]." points out all the weaknesses of the present paper. Semi-quantitative analysis requires a rigorous protocol, as all parameters must be reproducible and must follow standard procedures/guidelines. For this reason, this manuscript, while interesting and well-structured, should not be accepted in this form. 

Author Response

Referee 2

Comment 1

The paper addresses an interesting topic, as breast cancer, especially at an advanced stage, is a serious healthcare problem and monitoring response to treatment could provide fundamental support to clinicians in addressing the best treatment options and thereby improving patients' survival.

Reply: We thank the referee for this comment.

Comment 2

Nevertheless, in my opinion the sentence "Not all standardization criteria suggested by PERCIST could be documented due to the retrospective study design [12]." points out all the weaknesses of the present paper. Semi-quantitative analysis requires a rigorous protocol, as all parameters must be reproducible and must follow standard procedures/guidelines. For this reason, this manuscript, while interesting and well-structured, should not be accepted in this form. 

Reply: We acknowledge the limitations of this study due to the retrospective design as also provided in the limitation section. However, no studies have addressed the clinical utility of PERCIST for longitudinal response monitoring before and clinical validation seems of great importance.

The methods section has been improved with details on image techniques and standardization protocol to demonstrate that daily clinical practice in our instution to a large extent fulfills suggested standardization requirements by PERCIST and non-comparable scans were excluded.

We have elaborated on this in the discussion section. 

Marianne Vogsen

Round 2

Reviewer 2 Report

The manuscript is now improved and can be published on Diagnostics.

This manuscript is a resubmission of an earlier submission. The following is a list of the peer review reports and author responses from that submission.

Round 1

Reviewer 1 Report

  • Nice article, and the attention for a nadir as a reference value in the metastatic setting is of added value.
  • In the percist article it is adviced that 5 lesions are included in the analysis. In the introduction it is mentioned that in an article the one-lesion analysis delivered comparable results to the five-lesion analysis, but it is quite a shame that this was not analyzed as well.
  • Line 210: why were these pts not excluded from the analysis?
  • Line 213: that progression is found earlier when comparing a value to the lowest value compared to the baseline value is quite obvious.
  • Line 222: “than” is missing in the line. In several paragraphs of the article the english needs considerable improvement.
  • Line 259: this is too strong. It is unfortunate that it was not possible to check the standaridzation criteria of PERCIST. This is often an important limitation of the appicability of PERCIST, and this was not tested in this study.

Reviewer 2 Report

This study dealt with the PET response by PETCIST in metastatic breast cancer. Although this is an interesting topic, the lack of clinical values is major limitation of this study to consider the publication.

  1. In this study, visual response was used to compared with PECIST. However, it is necessary to describe the definition of visual response. Except PERCIST, the EORTC PET criteria (Young H, et al. Eur J Cancer 35(13):1773–1782) is usually used, which is recommended in this study.

  1. This study is a kind of descriptive study lack of the comparison with clinical outcomes. Because more time and effort are required for the PERCIST over conventional response evaluation, it is necessary to show the clinical values of PETCIST in metastatic breast cancer, however, which was not demonstrated in this study.

Reviewer 3 Report

This is a feasibility study of PERCIST (one lesion) compared to visual assessment in the response evaluation of metastatic breast cancer. The introduction of PERCISTnadir, as defined here, provides an interesting new dimension to the response assessment approach.

This is a retrospective analysis of a small number of patients with a different number of scans per patient; it utilises PERCIST in the longitudinal response assessment in metastatic breast cancer, an area which has not been studied so far.

The authors should address the following questions:

  1. One of the study limitations includes the lack of documentation on a number of PERCIST standardization criteria. To what extent do the authors expect this limitation to have influenced the results of the study? How was the interpretation of the data affected?
  2. The visual assessment of scans as used in the study is subject to multiple weaknesses owing to the retrospective nature of the study, also including different specialists reporting the scans. How did the authors minimize the interobserver variability of the visual assessment of scans before using this parameter as a comparator to the PERCIST?
  3. The load of metastatic disease in breast cancer may vary significantly from patient to patient, it would therefore be useful to present the sites and the burden of metastatic disease of the cases included in the study. Not all patients with metastatic breast cancer have widespread metastatic disease, unless the group of 37 cases used here had in similar load and spread of metastatic disease. If so please add to the text.
  4. Line 272-273: Comment on total lesion glycolysis (TLG): review in view of data requested in comment 3.
  5. Avoid repetition of information already presented in the Introduction Section (i.e. lines 267-272)